# Microwave a.c. conductivity of domain walls in ferroelectric thin films

Alexander Tselev[1], Pu Yu[2,3], Ye Cao[1], Liv R. Dedon[4], Lane W. Martin[4,5], Sergei V. Kalinin[1] & Petro Maksymovych[1]

Ferroelectric domain walls are of great interest as elementary building blocks for future electronic devices due to their intrinsic few-nanometre width, multifunctional properties and field-controlled topology. To realize the electronic functions, domain walls are required to be electrically conducting and addressable non-destructively. However, these properties have been elusive because conducting walls have to be electrically charged, which makes them unstable and uncommon in ferroelectric materials. Here we reveal that spontaneous and recorded domain walls in thin films of lead zirconate and bismuth ferrite exhibit large conductance at microwave frequencies despite being insulating at d.c. We explain this effect by morphological roughening of the walls and local charges induced by disorder with the overall charge neutrality. a.c. conduction is immune to large contact resistance enabling completely non-destructive walls read-out. This demonstrates a technological potential for harnessing a.c. conduction for oxide electronics and other materials with poor d.c. conduction, particularly at the nanoscale.

[1] Center for Nanophase Materials Sciences, Oak Ridge National Laboratory, Oak Ridge, Tennessee 37831, USA. [2] State Key Laboratory for Low-Dimensional Quantum Physics, Department of Physics and Collaborative Innovation Center for Quantum Matter, Tsinghua University, Beijing 100084, China. [3] RIKEN Center for Emergent Matter Science (CEMS), Wako, Saitama 351-0198, Japan. [4] Department of Materials Science and Engineering and Department of Physics, University of California, Berkeley, Berkeley, California 94720, USA. [5] Materials Science Division, Lawrence Berkeley National Laboratory, Berkeley, California 94720, USA. Correspondence and requests for materials should be addressed to A.T. (email: atselev@utk.edu) or to P.M. (email: maksymovychp@ornl.gov).

n ferroelectrics, domains of uniform polarizations are separated by domain walls. Domain walls can be created and reconfigured by electric fields[1,2], and their lateral width is only a few nanometre due to strong coupling between lattice strain and ferroelectric polarization[3,4]. In the past few years, the concept of domain wall electronics pursued utilization of domain walls[5,6] and other topological defects in ferroelectrics in view of applications in electronic devices[7,8]. Electronic conductivity is a basic example of the desired functionality. Despite electrically insulating nature of ferroelectric materials, several types of ferroelectric domain walls showed d.c. conduction[9–15]. However, the elucidation of the mechanism of the wall conduction and the progress towards practical utilization of domain wall circuits have so far been impeded by low conduction of domain walls and a large ferroelectric-electrode contact resistance[9–13,16]. Domain wall conductance was attributed to electronic reconstructions at the domain wall[9], segregation of mobile donor or charge-trapping defects[11,13], or finite charge of non-equilibrated domain wall morphology[1,10,12,14,17]. Electrostatically charged domain walls were proposed to be more conducting[14,15], but they are unstable and cannot be generally made without special conditioning[15], nanoscale topologies[18] or composition control[14].

The problem of contact resistance is particularly important in domain wall electronics. The contact interface Schottky barrier conceals the intrinsic mechanisms of charge transport along the in-depth region of a domain wall. Overcoming contact resistance necessitates electric fields that can become comparable or exceed the threshold for domain wall motion[10,11]. As a result, electronic measurements and prospective read-out of domain walls are generally destructive, capable of displacing and erasing the domain wall. At the same time, domain walls can reconstruct in an applied field, creating partially charged and more conducting configurations[10,11]. Domain wall reconstruction by itself can become practically useful provided there is a clear way to control domain morphology and analyse various conducting entities non-destructively[10,11], in a broad analogy to memristor electronics[19].

Here we reveal an alternative regime of ferroelectric domain wall conduction at a.c. frequencies in the gigahertz range. Measurements at high frequencies are insensitive to the contact barriers at the electrode–ferroelectric interface, enabling quantitative non-destructive measurement of the domain wall conductance. We found that nominally uncharged domain walls in conventional ferroelectric thin films of lead zirconate and bismuth ferrite can be strongly conducting, on par with charged walls. The a.c. conductivity is at least 100 times higher than at d.c. with the same probing voltage. From the behaviour of the a.c. domain wall conductance as a function of bias voltage, temperature and time, we conclude that the cloud of free carriers forming around the transient charged domain walls during polarization switching incompletely dissipates after the domain stabilization and partially remains at the wall. The domain wall assumes a rough morphology assisted by a disorder potential in the film. Such a configuration hinders d.c. conduction, but remains many fold more conducting at gigahertz frequencies than the surrounding domains, paving a way to practical utilization of domain wall circuits.

## Results

**Microwave microscopy of pristine films**. The experiments were carried out on a 100-nm-thick epitaxial thin film of $Pb(Zr_{0.2}Ti_{0.8})O_3$ (100) (PZT) grown using pulsed laser deposition on a single-crystal $TiO_2$-terminated $SrTiO_3$ (001) substrate with 50 nm $SrRuO_3$ as a bottom electrode (Fig. 1a). Hysteresis loops of remnant piezoresponse of the as-grown PZT film indicate

good ferroelectric properties of the film (Supplementary Fig. 1). The PZT film has a tetragonal lattice with the polarization axis along the surface normal. The as-grown ferroelectric polarization points to the film surface. Ferroelectric domains were created and manipulated using scanning probes, while a.c. conductivity was imaged and measured using a near-field scanning microwave impedance microscope (sMIM) operating at a frequency $f \approx 3$ GHz. Microwave microscopy has previously been shown to be capable of detecting conductance of the domain walls in bulk crystals[20,21]. As illustrated in Fig. 1a, the sMIM probe terminates a microwave transmission line. From the standpoint of the measurement system, the sample can be viewed as a capacitor $C_b$ and a resistor with conduction $G_b$ connected in parallel; the complex admittance of the tip–sample system $Y = G + i2\pi fC$ is measured, and real and imaginary parts are displayed in two channels: sMIM-$G$ and sMIM-$C$, respectively.

Figure 1b,c shows topography and piezoresponse force microscopy (PFM) images of a stripe domain structure formed by applying sequentially $V_{bias} = -7$ V and $+7$ V to the scanning probe. Ferroelectric polarization reverses between stripes separated by nominally 180° domain walls. Figure 1d shows a microwave conductance image (sMIM-$G$ channel) of the structure taken with a zero d.c. bias at the probe and an a.c. voltage amplitude $\lesssim 300$ mV. Domain walls are clearly conducting unlike the bulk of the surrounding domains, while no contrast between the two is seen in the simultaneous permittivity image (sMIM-$C$ channel; Supplementary Fig. 2). We have created domain structures of more complicated geometries that also showed microwave conductance of the domain walls. Erasing domain walls by reconfiguring ferroelectric domains leaves no observable traces in the sMIM-$G$ images. A 10-fold reduction of the a.c. voltage amplitude did not change the domain wall contrast (Supplementary Fig. 3). Imaging with a small-d.c. bias of $\sim 2$ V$_{d.c.}$ revealed oppositely polarized domains in the sMIM-$C$ channel (Supplementary Fig. 4a) due to dielectric tunability of PZT (which has opposite signs in oppositely polarized domains); however, the walls do not produce any contrast in the sMIM-$C$ image. In the corresponding sMIM-$G$ image (Supplementary Fig. 4b), domain walls showed approximately the same contrast as with the zero d.c. bias at the probe. Furthermore, the large a.c. conductance was detected with a film of $BiFeO_3$ (Supplementary Fig. 5) and several other PZT films, pointing to a general character of this phenomenon.

To gain insight into the mechanism of microwave conductance of the domain walls, we measured sMIM response as a function of d.c. bias at fixed locations on the sample surface. As displayed in Fig. 1e, d.c. bias sweeps between $-8$ and $+8$ V reveal dielectric tunability of PZT in the sMIM-$C$ channel with a curve shape typical for a ferroelectric. Kinks and jumps in the permittivity correspond to polarization switching under the probe. Meanwhile, the sMIM-$G$ channel (Fig. 1f) shows abrupt increase of conductivity at the switching events. The jumps can be identified as conductance due to tilted and charged domain walls of ferroelectric nanodomains, which were previously surmised to be electronically conducting due to accumulation of mobile carriers partially screening the bound charge along curved and tilted domain walls[18]. During domain growth, the a.c. conductance gradually decreases down to the bulk value, indicating that walls relax to the weakly charged state and eventually move out of the probed volume due to the growth of the domain in the d.c. electric field of the probe.

**Effect of annealing**. An increased a.c. conductance in the PZT film subject to heat treatment also supports that the a.c.

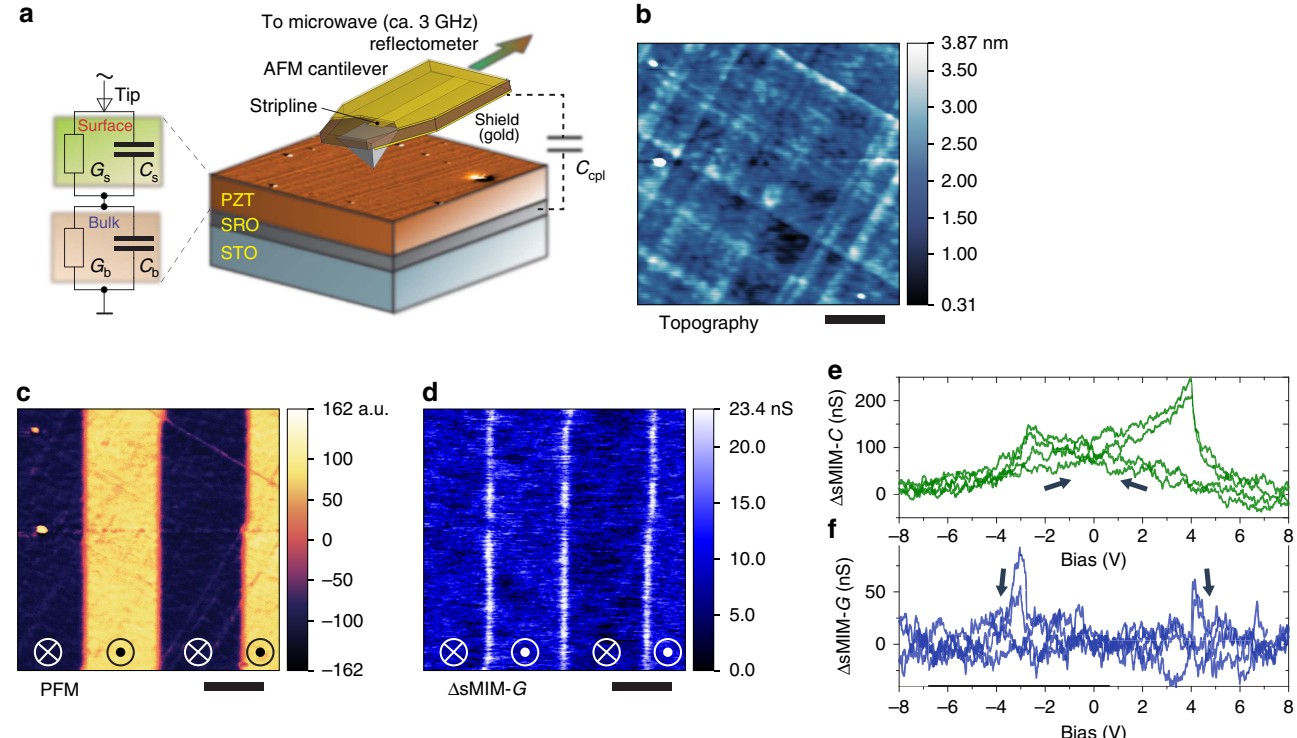

**Figure 1 | Microwave imaging of conducting domain walls. (a)** Schematic of the sample and sMIM microwave probe on an atomic force microscope (AFM) cantilever. An 100-nm-thick epitaxial thin film of Pb(Zr$_{0.2}$Ti$_{0.8}$)O$_3$ (PZT) with a 50-nm-thick SrRuO$_3$ (SRO) bottom electrode is deposited on an SrTiO$_3$(001) (STO) substrate. Microwaves of a frequency $f \approx 3$ GHz are delivered to the sensing tip of the probe and to a sample through a stripline fabricated on a silicon nitride AFM cantilever. A reflectometer measures amplitude and phase of the wave reflected from the tip, and represents the results as a change of the admittance $Y = G + i2\pi fC$ of the tip–sample system through two channels sMIM-$G$ and sMIM-$C$, corresponding to the conductance $G$ and capacitance $C$, respectively. Due to the capacitance of the space charge layer or conduction-blocking layers on the surface, electrical behaviour of the film surface can be described by a parallel resistor–capacitor circuit $G_s||C_s$ shown in the lumped elements diagram. The intrinsic dielectric response and conduction of the material bulk are represented by the pair $G_b||C_b$. Capacitor $C_{cpl}$ represents the large coupling capacitance between the SRO bottom electrode and the microwave probe shield. **(b)** Image of the surface topography. **(c)** Combined out-of-plane PFM image showing a stripe domain structure with polarization **P** orientated up ⊙ and down ⊗ in the ferroelectric domains as indicated in the image. **(d)** sMIM-$G$ image (that is, a map of the variation $\Delta$sMIM-$G$ of the reflectometer sMIM-$G$ signal) clearly reveals conductivity in the walls of the stripe domains seen in **b**. Images in **b–d** were obtained from the same area of the pristine PZT film. **(e)** Signal from the sMIM capacitance channel and **(f)** simultaneous signal from the sMIM conductance channel as functions of d.c. bias applied to the probe at a single point on the pristine film. Arrows indicate direction of the hysteresis. The signal in **f** is shown in respect to a reference signal approximately corresponding to the domain bulk. The images in **b–d** were acquired with a zero d.c. bias applied to the probe. Scale bars, 1 μm (**b–d**).

conduction of domain walls is associated with mobile charge carriers. Specifically, we annealed the PZT film under reducing conditions (in vacuum, $10^{-8}$ torr at 350 °C for 20 min). This depletes a small amount of oxygen and creates oxygen vacancies. The vacancies act as electron donors increasing the number of charge carriers in the film. The sMIM-$G$ response of the domain walls in the annealed film was ~1.5–2 times larger than the value of the as-grown film (compare Figs 1d and 2a,b). Noteworthy, annealing of the film also produced spontaneous a.c.-conducting walls (Fig. 2a). The magnitude of the conduction is comparable in spontaneous and recorded walls (Fig. 2c), as is the local sMIM response versus bias (Fig. 2g,h).

**d.c. conduction of the domain walls.** Both of the PZT films show d.c. conductance of the walls when probed by conductive atomic force microscopy (c-AFM; Fig. 3a,b). A rough estimate at a high bias (at 10 V) from the d.c. I–V curves for the domain bulk (Supplementary Figs 6 and 7) corresponds to ca. $2 \times 10^{-3}$ S m$^{-1}$ for the pristine film and ca. $5 \times 10^{-3}$ S m$^{-1}$ for the annealed film. In turn, to quantify the a.c. conductivity, we performed

calibrated measurements and numerical modelling of the tip–sample admittance alteration in the presence of a conducting domain wall (see the Methods section and Supplementary Figs 8 and 9 for details). The a.c. conductivity of the pristine film bulk was estimated 0.4–0.7 S m$^{-1}$ at 3 GHz. The domain wall a.c. conductivity fell in a range 4–8 S m$^{-1}$ assuming a 3-nm wall thickness, that is, ca. 10 times higher than in the bulk. At the same time, the d.c. conductivity is 100–200 times smaller than a.c. conductivity measured at 3 GHz, while a similar estimate for the domain wall in the pristine film (data of Fig. 3e) yields d.c. conductivity of ca. 0.1 S m$^{-1}$, that is, ~50 times lower than the a.c. value. We note that generally conductivity at gigahertz frequencies can be larger by orders of magnitude than at d.c. and show a relatively week, power-law, temperature dependence[22–28]. A fundamental reason behind this enhancement is that charge carriers localized by energy barriers at d.c. can contribute to a.c. conduction by oscillating between the barriers at high frequencies[22–24,26,28,29].

**Temperature dependence of the domain wall a.c. conduction.** In the PZT film, the a.c. conduction also showed negligible

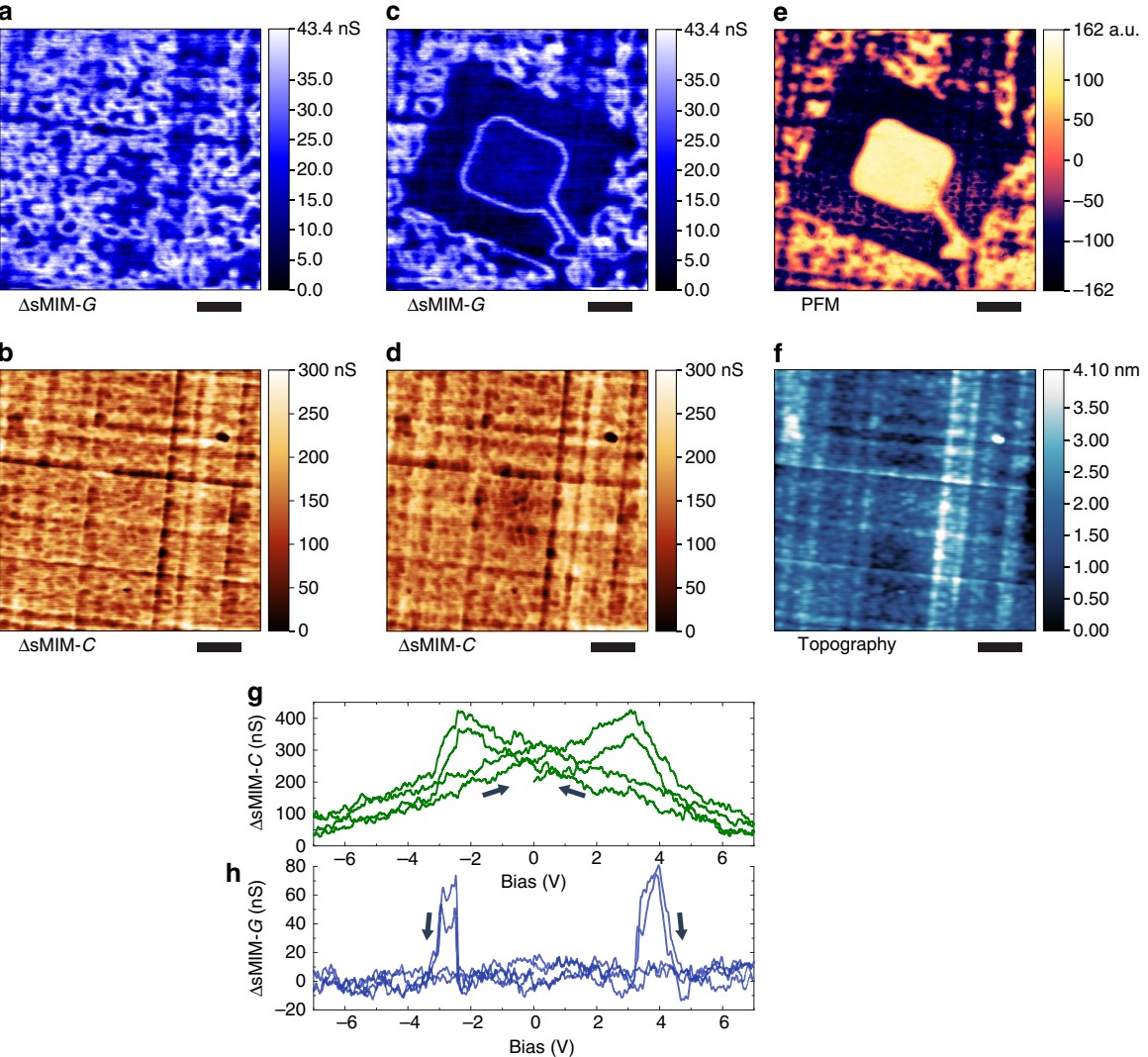

**Figure 2 | a.c. conductivity and manipulation of conducting domain walls in the annealed PZT film.** Images in **a**–**f** were obtained from the same area of the annealed film with a zero d.c. bias of the probe. (**a**) sMIM-*G* image, where conducting walls of spontaneous domains of a few-hundred-nm size are clearly seen. (**b**) sMIM-*C* image recorded simultaneously with the image in **a**. (**c**) sMIM-*G* image of the box-in-box domain structure written by the probe after the image in **a** was taken. Spontaneous domains and domain walls were erased inside the structure; the domain walls of the structure are conducting. (**d**) sMIM-*C* image recorded simultaneously with the image in **c**. (**e**) Combined out-of-plane PFM image acquired right after the image in **c**. (**f**) Image of the film surface topography recorded simultaneously with the image in **e**. Comparing images in **b**–**d** with the image in **f**, it is seen that the sMIM-*C* signal is dominated by a cross-talk with the surface topography. No traces of domain walls are seen in **d**. (**g**) Single-point signal from the sMIM capacitance channel and (**h**) simultaneous signal from the sMIM conductance channel as functions of d.c. bias applied to the probe for the annealed film. Arrows indicate the direction of the hysteresis. The signal in **h** is shown in respect to a reference signal approximately corresponding to the domain bulk. Scale bars,1 μm (**a**–**f**).

temperature dependence up to 115 °C (Fig. 4a–c). This is unlike d.c. conduction of domain walls, which could be both linear[14] and exponential[18]. Noteworthy as well, microwave conduction displays negligible dependence on d.c. bias away from switching events (Figs 1f, 2h and 4e). This is in stark contrast to rectifying d.c. *I*–*V* curves (Fig. 3e; Supplementary Figs 6 and 7), which are strongly non-linear, hysteretic, with current seen only at positive bias above a threshold of ∼1.3 V. Likewise, the images of the sMIM-*G* conductance weakly depend on the d.c. bias between −2 and 2 V. This behaviour of the a.c. response indicates no influence of the contact effects.

**Temporal stability**. The a.c. conductance of domain walls was stable over at least 48 h and showed no degradation. This time is sufficient for the 180° walls to equilibrate aligned along the polar

direction of the film. Such domain walls are nominally uncharged, and yet they show clear a.c. conduction, being minimally perturbed by the a.c. voltage (down to below 60 mV peak to peak at 3 GHz).

## Discussion

Whereas the a.c. conduction associated with mobile charge carriers oscillating between energy barriers explains the observed microwave response of the ferroelectric domain walls, domain wall vibrations near equilibrium positions forced by the high-frequency electric field of the probe should be considered as an alternative origin of the observed sMIM-*G* response to complete the picture. Such vibrations would result in microwave energy dissipation, which can be indistinguishable from the mobile charge high-frequency conduction in the sMIM measurements.

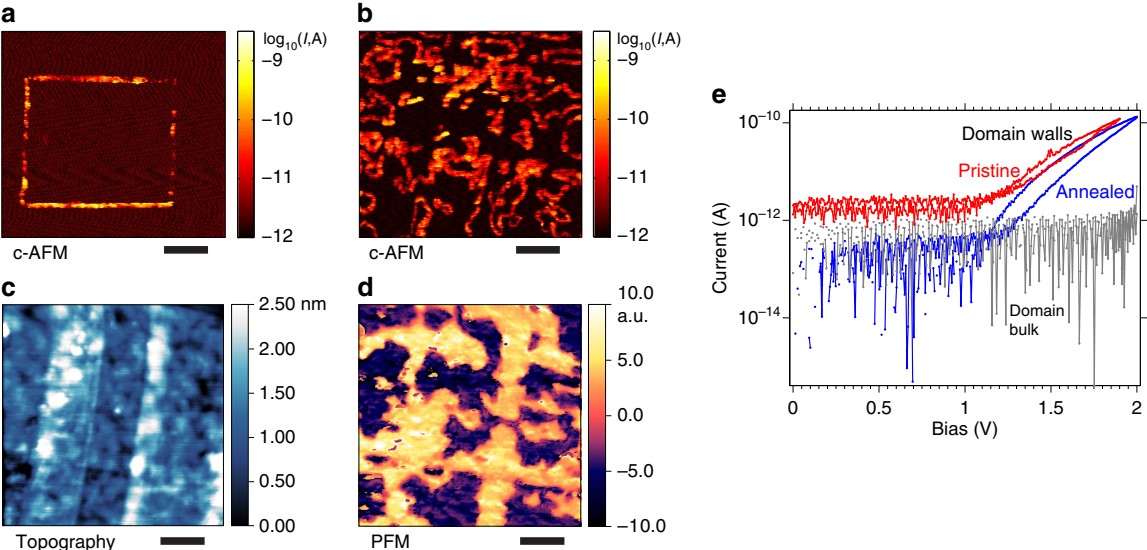

**Figure 3 | d.c. conduction of domain walls in PZT.** (**a**,**b**) Conductive AFM (c-AFM) images of the d.c. current, *I*, in the PZT film obtained with a metal-coated AFM probe in ultrahigh vacuum. Probe bias in **a** and **b** was fixed at +1.9 V. The image in **a** reveals a conducting domain wall written in the pristine film; the image in **b** shows a spontaneously formed network of conducting domain walls in the annealed film. (**c**,**d**) Image of surface topography and a combined out-of-plane PFM image, respectively, acquired simultaneously from the annealed film. The spots on the film, where images in **b**–**d** were recorded are different, but located close to each other. (**e**) Probe current versus bias for wall locations and for the domain bulk away from walls. Red, blue and grey correspond, respectively, to a domain wall in the pristine film, a domain wall in the annealed film and domain bulk (which does not show detectable conduction at d.c. both in the pristine and annealed films). Scale bars, 280 nm (**a**); 400 nm (**b**–**d**).

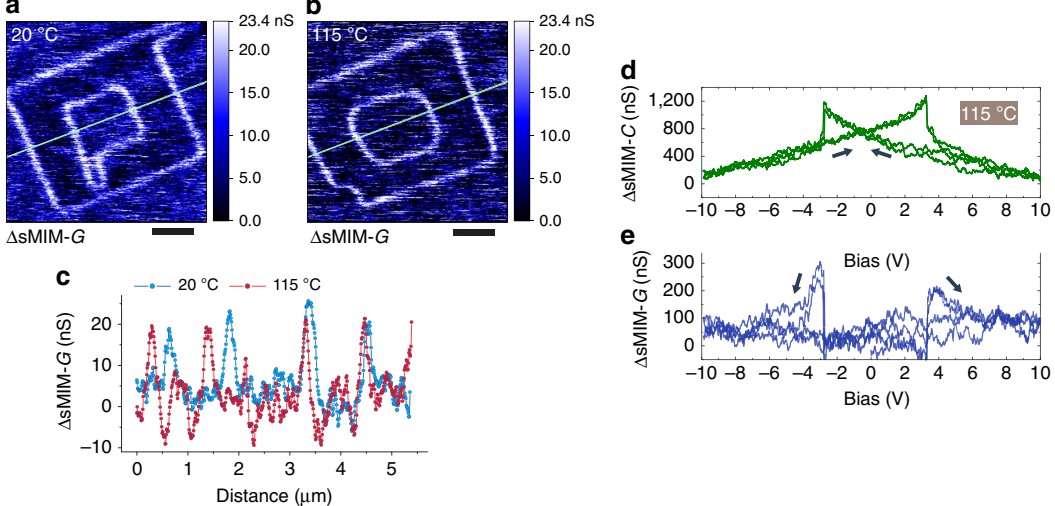

**Figure 4 | Temperature dependence of the a.c. conduction of domain walls.** (**a**,**b**) sMIM-*G* images of box-in-box domain structures written and imaged at different sample temperatures. The images were acquired with a zero d.c. bias of the probe. The image in **a** was obtained at room temperature, and the image in **b** was obtained at a sample temperature of 115 °C. As seen, the intensity of the domain wall response in respect to the background is the same for the images in **a** and **b**. This is further illustrated in **c**. (**c**) Signal profiles along the green lines in images (**a**,**b**). Blue and red correspond to room temperature and 115 °C, respectively. The curves were slightly offset along the vertical axis for the ease of comparison. Note that the geometry of the box-in-box structure is distorted in **b** due to thermal drift. (**d**) Single-point signal from the sMIM capacitance channel and (**e**) simultaneous signal from the sMIM conductance channel at a sample temperature of 115 °C as functions of d.c. bias applied to the probe for the annealed film. Arrows indicate the direction of the hysteresis. The signal in **e** is shown in respect to a reference signal approximately corresponding to the domain bulk. Signals in **d** and **e** are stronger than in other experiments described in the paper because of an increased sample–probe force and, consequently, a larger sample–probe contact area. This was done to increase signal-to-noise ratio to compensate for the increase of the noise level associated with the elevated temperature of the probe. Scale bars, 1 µm (**a**,**b**).

However, it should be taken into account that displacement of domain walls in response to the applied electric field contributes to material polarizability and permittivity[30]. When a domain wall is present under the sMIM probe, it is expected that the extrinsic contribution of the domain wall vibration to the intrinsic material permittivity is significant and, therefore, can be detected in the capacitance channel if the associated energy loss is detectable by the conduction channel. In our experiments, however, while changes of the film permittivity could be readily observed by tuning the permittivity with an applied bias, no contrast from conducting domain walls could be seen in the corresponding sMIM-C images (Fig. 2d; Supplementary Figs 2 and 4). This is

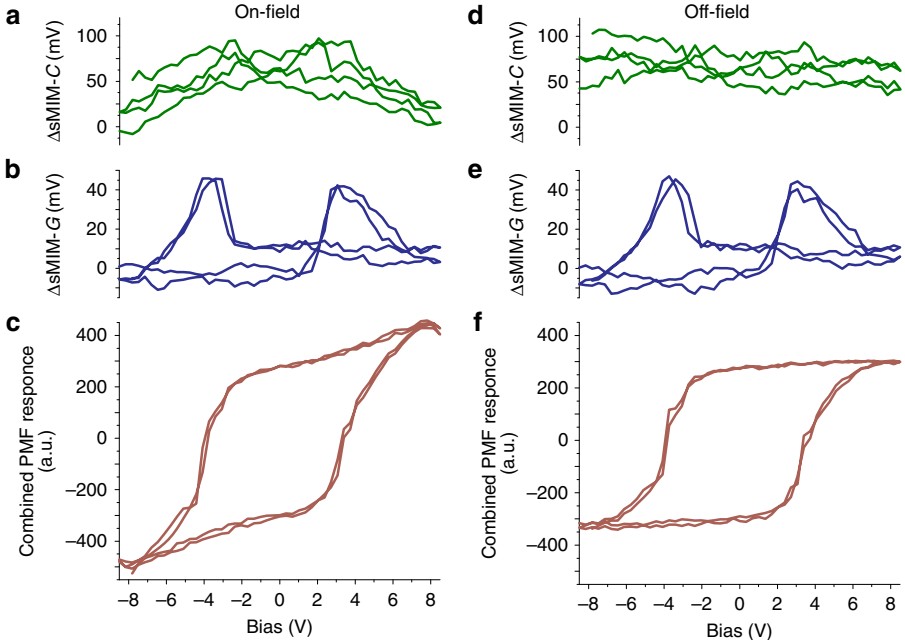

**Figure 5 | Single-point sMIM measurements simultaneous with switching spectroscopy PFM.** (**a**) On-field sMIM-*C* and (**b**) on-field sMIM-*G* signals corresponding to (**c**) on-field out-of-plane PFM hysteresis loop for a BiFeO₃ film. (**d**) Off-field sMIM-*C* and (**e**) off-field sMIM-*G* signals corresponding to (**f**) remnant out-of-plane PFM hysteresis loop of the film. The sMIM signals are uncalibrated.

further strengthened by single-point sMIM measurements combined with simultaneous switching spectroscopy PFM. In an switching spectroscopy PFM experiment, voltage bias is applied to the probe in a series of pulses of varying amplitude, and PFM as well as sMIM responses are measured during the application of a pulse and in between consecutive pulses (when the bias is set to zero)[31]. The responses during pulse application (on field) and in between corresponding pulses (off field or remnant) are plotted separately as functions of the pulse amplitude. As illustrated in Fig. 5 for a film of BiFeO₃ as an example, while the on-field sMIM-*C* signal (Fig. 5a) shows the trend expected for the dielectric tunability, the remnant sMIM-*C* response (Fig. 5d) is constant over probe bias, being the same at the polarization switching (when domain walls are present under the probe) and away from it (when the probe is surrounded by a uniformly polarized material). At the same time, the on- and off-field sMIM-*G* signals (Fig. 5b,e) are nearly identical, with conduction peaks around switching evens. A very similar behaviour was observed with PZT and BiFeO₃ films, evidencing that the effect is weakly dependent on the specific nature of the material, and hence surrounding and dynamic characteristics of the domain walls. It can be concluded that the domain wall vibrations, if present, do not contribute enough to be detectable and to explain the observed a.c. conduction of domain walls.

To explain the large a.c. conduction of domain walls, we note that domain wall pinning by lattice defects, and associated strain and field disorder will disrupt the idealized straight shape of the wall making it locally curved[32–34]. The curvature in respect to polarization will translate into bound charges distributed along the roughened domain wall and compensated by localized clouds of mobile carriers, which are responsible for the enhanced a.c. conductivity. This effect is reminiscent of a.c. conduction in metal–insulator composites with metal concentrations below the critical value for percolation threshold[24]. To model this effect, we implemented phase-field modelling[35] in the presence of random field disorder (see the Methods section for details). As seen in Fig. 6a–c, the disorder indeed significantly roughens the

otherwise smooth domain wall, creating local head-to-head and tail-to-tail polarization configurations along the 180° walls, which is revealed in the modelled electric potential distribution map in Fig. 6d. Above a certain disorder strength, substantial electron accumulation is observed along the domain wall (Fig. 6e,f).

In conclusion, we revealed a large a.c. conductance of nominally uncharged ferroelectric domain walls in common oxide ferroelectrics. We explain the a.c. conduction by the domain wall roughness associated with disorder in the ferroelectric films. Taking into account the universal applicability of the a.c. conduction, implications of our findings can be extended to other materials, especially where the a.c. conductivity can be controlled with external fields. This may lead to the emergence of novel device paradigms based on a large spectrum of functionalities potentially offered by complex oxide and nanoscale material systems.

## Methods

**Pb(Zr₀.₂Ti₀.₈)O₃ film sample fabrication.** The ferroelectric Pb(Zr₀.₂Ti₀.₈)O₃ (100) thin film was grown using pulsed laser deposition on a single-crystal TiO₂-terminated SrTiO₃ (001) substrate with 50 nm SrRuO₃ as a bottom electrode. The growth temperatures were 700 and 630 °C for SrRuO₃ and PZT layers, respectively. The growth oxygen pressure was the same for both materials— 0.13 mbar. After growth, the sample was cooled down at 1 bar oxygen with a cooling rate of 5 °C min⁻¹. The thickness of the PZT film was controlled by the growth rate and was confirmed with X-ray reflectometry.

**BiFeO₃ film sample fabrication.** The 100 nm BiFeO₃/50 nm SrRuO₃/DyScO₃ (110) heterostructure was grown using pulsed laser deposition from a Bi₁.₁FeO₃ target. The SrRuO₃ bottom electrode was grown at 645 °C in a dynamic oxygen pressure of 100 mtorr at a laser fluence 1.8 J cm⁻² and a frequency of 17 Hz. The BiFeO₃ film was grown at 700 °C in a dynamic oxygen pressure of 100 mtorr at a laser fluence of 1.0 J cm⁻² and a frequency of 20 Hz. Following growth, the heterostructure was cooled in a static oxygen pressure of 760 torr at a rate of 5 °C min⁻¹.

**PFM and c-AFM imaging.** PFM imaging in ambient was performed in the same set-up and with the same probes as the sMIM imaging (see below, the Microwave imaging and measurements section) using the Asylum Research Dual AC Resonance Tracking mode. A 1.5-V_PP a.c. voltage at a frequency close to the contact

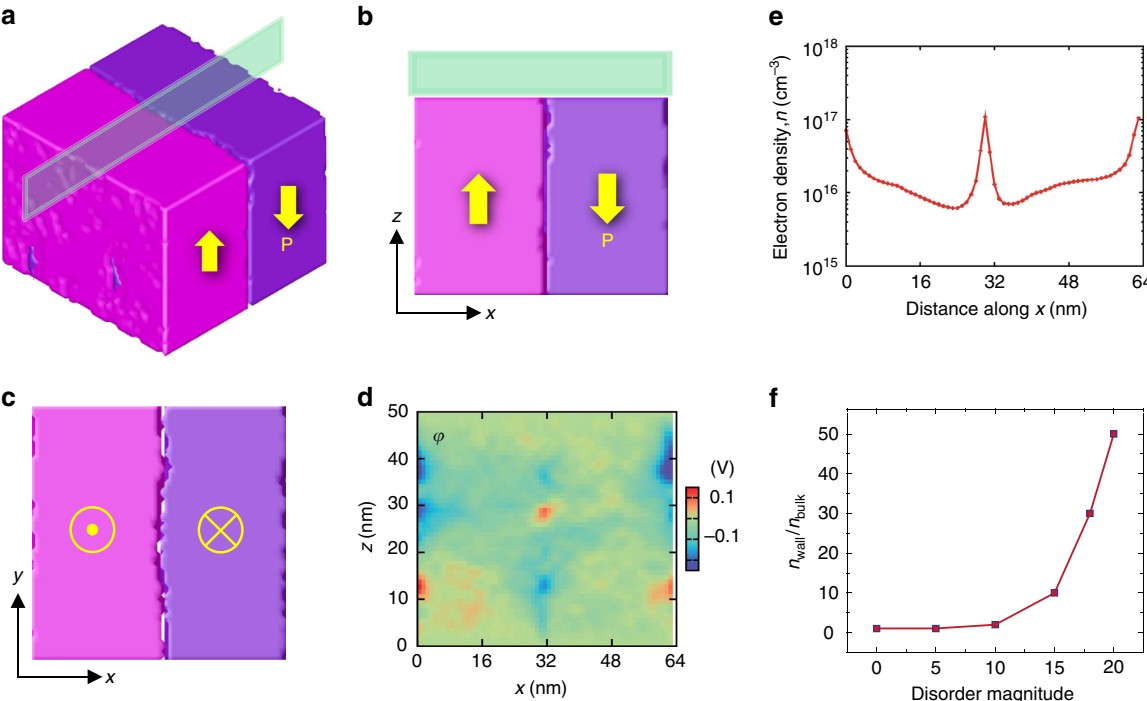

**Figure 6 | Phase-field simulation of the ferroelectric domain structure.** The simulations were performed for a $Pb(Zr_{0.2}Ti_{0.8})O_3$ film with a uniform defect disorder. (**a**) Equilibrium three-dimensional domain structure with rough 180° domain walls. Parallel domain walls are in the centre (visible in the figure) as well as at the sides due to periodic boundary conditions. (**b,c**) Two-dimensional plots of domain structure in the model $x$–$z$ plane at $y = n_y/2$ (indicated by a green rectangle in **a**) and on the top surface ($x$–$y$ plane), respectively. ($n_y$ is the model size along the $y$ direction.) The effect of the random field on the wall roughness is less obvious in the film interior than in the surface vicinity. (**d**) Electric potential distribution in the $x$–$z$ plane at $y = n_y/2$. The electric potential reaches 0.1 and $-0.2$ V in the wall region. (**e**) Electron density averaged along the normal to the film surface ($z$ direction) throughout the film thickness as a function of the position perpendicular to the walls ($x$ direction). The plot shows a higher electron density in the wall vicinity than in the domain bulk. (**f**) Averaged ratio of electron density in the wall and bulk regions versus disorder magnitude. The disorder magnitude was set to $M = 15$ for calculation of the data for the curve in **e** and for the map in **d**.

resonance frequency of the probe ($\sim$250 kHz) was applied for imaging. The c-AFM and corresponding PFM imaging were performed in an Omicron VT AFM in ultrahigh vacuum at a background pressure of $2 \times 10^{-10}$ torr using Budget Sensors ElectriMulti75-G cantilevers.

**sMIM imaging and signal calibration.** Microwave imaging and measurements were performed in ambient with a ScanWave (Prime Nano, Inc.) sMIM add-on unit installed on an Asylum Research MFP-3D atomic force microscope. The sMIM microwave output power was set to 100 μW ($-10$ dBm), at which the upper estimate for the a.c. amplitude at the probe tip is ca. 300 mV$_{a.c.}$. Fully shielded sMIM cantilever probes (Prime Nano, Inc., see also ref. 36) had spring constants in a range 7–8 N m$^{-1}$. The probes allow both microwave and PFM imaging. The sensing pyramid of the sMIM probes is made of Ti/W alloy, and this was used to increase the sensitivity of sMIM imaging at the expense of spatial resolution. Namely, the set point of the microscope was increased for a short time by a factor of 3–5 in comparison with normal imaging regime to increase the tip–sample loading force. Effectively, this increased the tip–sample contact area, and increased the probe–sample capacitive coupling.

To correctly define the sMIM-C and -G channels, the ScanWave's detector phase was adjusted while imaging a set of 1-μm diameter capacitors of a SMM calibration kit from MC2 Technologies (France). The phase was adjusted so that the variation $\Delta$sMIM-$G$ of the reflectometer sMIM-$G$ signal from the capacitors was zeroed. Supplementary Fig. 10 illustrates sensitivity of the image contrast to the detection phase setting.

The sMIM-$C$ channel sensitivity to admittance changes was calibrated using an MC2 Technologies SMM calibration kit based on the approach of ref. 37. The calibrated sensitivity value is applicable to the sMIM-$G$ channel as well and can be used to measure the real part of the admittance of the tip–sample system on contact with a sample. The microscope sensitivity was determined as $\Delta V_{out}/\Delta Re,Im(Y)$, where $\Delta V_{out}$ is the raw, uncalibrated, voltage outputs at sMIM channels (in mV), and $\Delta Re,Im(Y)$ are the corresponding changes of the tip–sample admittance real and imaginary parts (in nS).

**sMIM measurements of permittivity and conductivity.** The PZT relative permittivity $\varepsilon_{PZT}$ was measured using the sMIM-$C$ microscope response in respect

to air (probe is out of contact with a sample) as a reference signal and an MC2 calibrations kit. The measurements yield $\varepsilon_{PZT} \approx 70 \pm 5$, which is in a good agreement with the literature value 70–100 for the same composition[38]. The conductivity of the domain bulk was determined from the sMIM-$G$ signal change in respect to air. Since the sMIM outputs depend on the probe–sample contact area, the contact diameter was measured using the apparent domain wall thickness in PFM images. The real part of the admittance over domain walls was measured from domain wall images based on the sMIM-$G$ signal change in respect to the domain bulk. The data for measurements were taken from images of the domain walls obtained after writing. While creating a suitable domains structure, a special attention was paid to stability of tip–sample contact through all writing sequence to preserve the validity of the probe calibration. Quantifying measurements for the film bulk were carried out with larger tip–sample contact areas to ensure a sufficiently large signal-to-noise ratio. The calibration measurements and measurements of the tip–sample contact diameter were performed before and after each image intended for quantification of permittivity and/or conductivity.

**Measurement of the probe–sample contact size.** PFM images of the domain walls were used to determine the tip–sample contact diameter. Since the domain wall thickness is 1–2 nm, that is, much smaller than the contact diameter, the apparent width of the domain walls in the PFM images can be taken as the tip–sample contact diameter with a high accuracy. This method was used for quantification of the film conductivity from both sMIM and c-AMF data. In sMIM measurements, the contact diameter was 100–350 nm, depending on the probe wear. In turn, in c-AFM measurements, the contact diameter was 25–30 nm.

**sMIM imaging at elevated temperatures.** For imaging at elevated temperatures, the sample was glued to a heater with a silver paint. Sample temperature was measured with a miniature thermocouple attached to the sample. The sample temperature of 115 °C was maximal, when properties of the sMIM probe remained stable and imaging was possible. For better thermal stability, the experiments were started following a 10-min waiting period after the temperature of 115 °C was reached. Because heating of a sample warms up the probe as well, the phase setting of the sMIM electronics was adjusted at 115 °C after that. The probe was kept retracted from the sample surface by 3–5 μm during heating and waiting periods.

To avoid tip-apex ware and contamination (to ensure the tip–sample contact stability for a valid comparison of the results), we did not perform a search of the structure written at room temperature after a temperature point was reached, but prepared another similar structure. The domain structures were prepared and imaged at electrically identical conditions at room temperature and at 115 °C.

**Numerical (finite elements) modelling.** The calibrated measurement values were applied to quantify the conductivities of the PZT film bulk and domain walls via numerical modelling. Numerical modelling for a frequency $f = 3$ GHz was performed using an a.c./d.c. module of the COMSOL v.4.2a Multiphysics finite elements analysis package (COMSOL AB). Two-dimensional axisymmetric models were used in the simulations. The conductivity of the SrRuO$_3$ film was set to $10^5$ S m$^{-1}$ (ref. 39). The domain wall in the model (Supplementary Fig. 8) was represented by a cylinder placed coaxial with the tip. The cylinder radius was set so that the contact area between the tip and cylinder was equal to the area of the straight-wall cross-section along the tip–sample contact diameter. Taking the wall thickness equal to 3 nm and tip–sample contact radius equal to 160 nm, this area is 660 nm$^2$. The model does not account for the surface roughness, which may reduce the contact area in the experiments.

In the simulations, first, the PZT film permittivity was set to a certain value in a range $\varepsilon_{\text{PZT}} = 70 \pm 5$, and conductivity of the uniform PZT film was swept to find a conductivity matching the measured value of the sMIM-$G$ response for the domain bulk. Then, the herewith determined conductivity for the film was fixed, while the domain wall conductivity was swept to match an experimental value of the tip–sample conductance over a domain wall. With $\varepsilon_{\text{PZT}} = 70$ and the other parameter values as listed above, the a.c. conductivities of the PZT domain bulk and domain wall were determined to be $\sim 0.6$ and 6 S m$^{-1}$, respectively, for the pristine film. The ranges 0.4–0.7 S m$^{-1}$ for the pristine film bulk and 4–8 S m$^{-1}$ for the domain wall, as listed in the main text, take into account uncertainty of the $\varepsilon_{\text{PZT}}$ measurement as well as uncertainties of the sMIM-$G$ signal due to the measurement noise and instabilities in the tip–sample contact.

**Phase-field simulations.** The spatial and temporal evolution of Pb(Zr$_{0.2}$Ti$_{0.8}$)O$_3$ domain structure was simulated using a phase-field model by numerically solving the time-dependent Landau–Ginzburg–Devonshire equations[35]:

$$\frac{\partial P_i}{\partial t} = -L\frac{\delta F}{\delta P_i}\,(i = 1, 2, 3), \qquad (1)$$

where **P** is the ferroelectric polarization vector, $L$ is a kinetic coefficient related to the domain motion, $t$ is time and $F$ is the total free energy, which is expanded as:

$$F = \int_V \left[ f_{\text{bulk}}(P_i) + f_{\text{grad}}(\nabla P_i) + f_{\text{elas}}(P_i, \varepsilon_{kl}) + f_{\text{elec}}(P_i, E_i) \right] dV, \qquad (2)$$

where $f_{\text{bulk}}(P_i)$, $f_{\text{grad}}(\nabla P_i)$, $f_{\text{elas}}(P_i, \varepsilon_{kl})$ and $f_{\text{elec}}(P_i, E_i)$ represent the Landau–Ginzburg–Devonshire free energy density of the uniform strain-free bulk, the gradient energy density, the elastic energy density and the electrostatic energy density, respectively. Details of the expressions for each of the energy density terms are summarized in refs 40,41. Without loss of generality, to investigate the role of the defect disorder on the 180° domain walls, we introduce disorder as a fixed random built-in electric field $\mathbf{E}_{\text{bi}}(x,y,z)$, which can be associated with charged defects randomly distributed in the film volume. The $x$, $y$ and $z$ components of $\mathbf{E}_{\text{bi}}$ at a point in the film volume are:

$$\begin{aligned} E_{\text{bi}-x} &= A\cos(\alpha)\sin(\beta), \\ E_{\text{bi}-y} &= A\cos(\alpha)\cos(\beta), \\ E_{\text{bi}-z} &= A\sin(\alpha), \end{aligned} \qquad (3)$$

where $\alpha$ and $\beta$ describe the random field orientation and are uniformly distributed random numbers between $-2\pi$ and $2\pi$. $A$ is the amplitude of the random field $A = M \cdot f(\mu)$, in which $M$ is the disorder magnitude and $f(\mu)$ is the Gaussian function with $\mu$ being a uniformly distributed random number between $-1$ and 1. To determine the local electric potential and the spatial distribution of free charge carrier in the presence of the charged defects, coupled equations:

$$n = N_c F_{1/2}\left(\frac{E_f - E_c + q\varphi}{k_B T}\right), \qquad (4)$$

$$p = N_v F_{1/2}\left(\frac{E_v - E_f - q\varphi}{k_B T}\right), \qquad (5)$$

$$-\nabla^2\varphi = \frac{q \cdot z_i(N_d + p - n - N_a) - \nabla \cdot P_i}{\varepsilon_0 \varepsilon_r} \qquad (6)$$

are solved, where $N_d$, $N_a$, $n$ and $p$ denote the local concentrations of donors, acceptors, electrons and holes, respectively. $N_c$ and $N_v$ are the effective density of states of electrons in the conduction and holes in the valance bands, respectively ($\sim 10^{21}$ cm$^{-3}$; ref. 42). $E_c$, $E_v$ and $E_f$ are the energies of the conduction band edge, valance band edge and Fermi level of PZT, respectively, $q$ is the unit charge and $\varphi$ is the electric potential. $F_{1/2}$ is the Fermi–Dirac integral, $k_B$ is the Boltzmann constant, $T$ is the absolute temperature, $z_i$ is the charge number of each charged

species, and $\varepsilon_0$ and $\varepsilon_r$ are the vacuum permittivity and the relative permittivity of PZT, respectively.

The PZT film is assumed to be heavily n-doped. Equation (1) is solved using a semi-implicit spectral method[43] with periodic boundary conditions along $x$, $y$ and $z$ directions. The simulation volume is discretized into a three-dimensional mesh $64\Delta x \times 64\Delta x \times 64\Delta x$, in which $\Delta x$ is set to 1 nm. The thicknesses of the film and substrate are assumed to be $50\Delta x$ and $10\Delta x$, respectively. The in-plane epitaxial compressive strain is chosen to be 1.0%. The energies of the conduction band edge, valance band edge and the Fermi level are set to $-4.0$, $-7.4$ and $-4.2$ eV, respectively. The PZT relative permittivity is assumed to be 50. The Landau coefficients, electrostrictive coefficients and elastic compliance constants of PZT have been found in refs 44–46.

**Data availability.** The data that support the findings of this study are available from the corresponding authors on request.

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

## Acknowledgements

We thank Stuart Friedman for technical support and Ramamoorthy Ramesh for assistance with film growth, and Long-Qing Chen for input in the phase-field model. This research was sponsored by the Division of Materials Sciences and Engineering, Office of Science, Basic Energy Sciences, US Department of Energy (A.T., S.V.K. and P.M.). Scanning probe measurements were conducted at the Center for Nanophase Materials Sciences, which is sponsored at Oak Ridge National Laboratory by the Scientific User Facilities Division, Office of Basic Energy Sciences, US Department of Energy. P.Y. was financially supported by the National Basic Research Program of China (grant 2015CB921700) and National Natural Science Foundation of China (grant 11274194). L.R.D. and L.W.M. acknowledge support from the Office of Basic Energy Sciences, US Department of Energy under grant no. DE-SC0012375.

## Author contributions

A.T. and P.M. conceived the experiments and the model, performed measurements and wrote the paper. P.Y., L.R.D. and L.W.M. fabricated thin-film samples. Y.C. performed phase-field modelling. All authors discussed the results and commented on the manuscript.

## Additional information

**Competing financial interests:** ORNL filed provisional US patent application based on the described technology.

