## [Peer Review File · Nature Communications]

REVIEWERS' COMMENTS:

Reviewer #1 (Remarks to the Author):

The manuscript presents scanning Microwave Impedance Microscopy (sMIM) of domain walls (DW) in ferroelectric thin films. Surprisingly high and stable ac conductivity is reported at DWs and its explanation is based on a newly added phase field simulation which considers random-field disorder that causes higher carrier concentration at DWs due to a zig-zag profile of DWs.

The manuscript went through certain scrutiny and authors answered the reviewers questions satisfactorily in my opinion. Although the claims about magnitude of the ac conductivity were reduced to less impressive DW/bulk ratio ~ 10 , and the interpretation required new assumptions - the DW roughness, it still represents an intriguing result which I recommend for publication in Nature Communications as is with one recommendation.

I recommend to discuss following in the manuscript unless the authors honestly justify (to themselves) that it is irrelevant. The sMIM measures microwave reflectance due to the impedance matching between the probe and the measured object. Small surface features obviously affect the measured reflectance which is probably unrelated to the conductivity. I wonder whether it can be disqualified that the measured signal comes from impedance mismatch due to the up-down change of the spontaneous polarization and not due to the free carriers at the DW? This cannot be really calibrated. If the signal is a result of some polarization wiggling at the DW/surface intersection, could any signal be transferred through the DW?

Reviewer #2 (Remarks to the Author):

With their revised version the authors present a detailed response to the remarks of both reviewers. They present additional evidence on the involvement of oxygen vacancies and they identify effects related to a certain roughness of the domain walls as additional contributing mechanism to the impedance. This still leaves a big question mark as to the exact microscopic mechanisms of the large impedance effect at the domain walls but including this would lead beyond the scope of this work. I thus recommend the manuscript for publication in Nature Communications. Since there is no length limit here, the authors might consider to shift some of the materials presented as supplementary information to the main text.

Response to reviewers' comments

We thank reviewers for careful reading of our manuscript and for the high appreciation of our work. Below, please find our detailed responses to the reviewers' remarks.

Reviewer #1 (Remarks to the Author):

The manuscript presents scanning Microwave Impedance Microscopy (sMIM) of domain walls (DW) in ferroelectric thin films. Surprisingly high and stable ac conductivity is reported at DWs and its explanation is based on a newly added phase field simulation which considers random-field disorder that causes higher carrier concentration at DWs due to a zig-zag profile of DWs.

The manuscript went through certain scrutiny and authors answered the reviewers questions satisfactorily in my opinion. Although the claims about magnitude of the ac conductivity were reduced to less impressive DW/bulk ratio ~ 10 , and the interpretation required new assumptions - the DW roughness, it still represents an intriguing result which I recommend for publication in Nature Communications as is with one recommendation.

I recommend to discuss following in the manuscript unless the authors honestly justify (to themselves) that it is irrelevant. The sMIM measures microwave reflectance due to the impedance matching between the probe and the measured object. Small surface features obviously affect the measured reflectance which is probably unrelated to the conductivity. I wonder whether it can be disqualified that the measured signal comes from impedance mismatch due to the up-down change of the spontaneous polarization and not due to the free carriers at the DW? This cannot be really calibrated. If the signal is a result of some polarization wiggling at the DW/surface intersection, could any signal be transferred through the DW?

Response. We appreciate the reviewer's suggestion. This is indeed a possible alternative explanation, which, however, should be ruled out based on our data. We would like to note as well that a very similar question was raised by Reviewer #2 and answered in our response in the previous round of the peer-review communications. Given more space available, we have now included a paragraph in the main text discussing why domain wall vibrations cannot be employed to explain our observations (the first paragraph in Discussion section of the revised version). The paragraph is accompanied by a new Fig. 5. (The previous Fig. 5 became Fig. 6 now.) Here is the text of the new paragraph:

“Whereas the ac conduction associated with mobile charge carriers oscillating between energy barriers explains the observed microwave response of the ferroelectric domain walls, domain wall vibrations near equilibrium positions forced by the high-frequency electric field of the probe should be considered as an alternative origin of the observed sMIM-G response to complete the picture. Such vibrations would result in microwave energy dissipation, which can be indistinguishable from the mobile-charge high-frequency conduction in the sMIM measurements. However, it should be taken into account that displacement of domain walls in response to the applied electric field contributes to material polarizability and permittivity²⁹. When a domain wall is present under the

sMIM probe, it is expected that the extrinsic contribution of the domain wall vibration to the intrinsic material permittivity is significant and, therefore, can be detected in the capacitance channel if the associated energy loss is detectable by the conduction channel. In our experiments, however, while changes of the film permittivity could be readily observed by tuning the permittivity with an applied bias, no contrast from conducting domain walls could be seen in the corresponding sMIM-C images (Fig. 2d, Supplementary Fig. 2 and 4). This is further strengthened by single-point sMIM measurements combined with simultaneous switching spectroscopy PFM (SS-PFM). In an SS-PFM experiment, voltage bias is applied to the probe in a series of pulses of varying amplitude, and PFM as well as sMIM responses are measured during application of a pulse and in between consecutive pulses (when the bias is set to zero)³⁰. The responses during pulse application (on-field) and in between corresponding pulses (off-field, or remnant) are plotted separately as functions of the pulse amplitude. As illustrated by Fig. 5 for a film of BiFeO₃ as an example, while the on-field sMIM-C signal (Fig. 5a) shows the trend expected for the dielectric tunability, the remnant sMIM-C response (Fig. 5d) is constant over probe bias, being the same at the polarization switching (when domain walls are present under the probe) and away from it (when the probe is surrounded by a uniformly polarized material). At the same time, the on- and off-field sMIM-G signals (Fig. 5b,e) are nearly identical with conduction peaks around switching events. A very similar behavior was observed with PZT and BiFeO₃ films evidencing that the effect is weakly dependent on the specific nature of the material and, hence, surrounding and dynamic characteristics of the domain walls. It can be concluded that the domain wall vibrations, if present, do not contribute enough to be detectable and to explain the observed ac conduction of domain walls.”

We have also put some more stress on the discussion of no-contrast between domain walls and domain bulk in sMIM-C images by adding the following text in the second paragraph of the Results section, ‘Microwave microscopy of pristine films’ subsection: “Imaging with a small dc bias of about 2 V_{dc} revealed oppositely polarized domains in the sMIM-C channel (Supplementary Fig. 4a) due to dielectric tunability of PZT (which has opposite signs in oppositely polarized domains); however, the walls do not produce any contrast in the sMIM-C image. In the corresponding sMIM-G image (Supplementary Fig. 4b), domain walls showed approximately the same contrast as with the zero dc bias at the probe.”

Immediately below, we reproduce for your convenience the question by Reviewer #2 and our response from the first round of the peer-review (figures are numbered as in the previous version of the manuscript and Supplementary information).

Reviewer #2. *I do not see many fundamental issues which could be questioned in this work. The main which I find is whether the conductance and susceptance can be distinguished with so high confidence. I suppose the reflectometry-based impedance measurement is strongly phase sensitive. It is rather surprising that DWs show almost no change in susceptance. DWs should be the places where bound charges lie in a very flat energy minimum which implies high permittivity. Therefore I wonder how reliable is the phase detection of the measured admittance. Could an error be evaluated or discussed?*

Response. To show that conductance in the sMIM-G channel can be reliably distinguished from the change in the dielectric constant in the sMIM-C channel, we performed imaging with accurately

adjusted detection phase and the phase set 3° off the correct value. The result of this experiment is shown in Supplementary Fig. 13. It is seen that the small change in the phase setting does not lead to significant degradation of the contrast from the domain walls in the sMIM-G channel. It results, however, in the “leakage” of the sMIM-G signal into sMIM-C channel. We also note that the sMIM-C image reveals a weak contrast from the domain bulk due to a small difference of the dielectric constants of up and down domains (please see Supplementary Fig. 14 and associated text in Supplementary Methods, which provide a clarification of this contrast). However, the domain walls do not appear in the sMIM-C images.

We do fully agree that the domain wall vibrations near the equilibrium positions should cause a Debye-like dispersion of the dielectric response at the domain walls with downturn of the real part of permittivity at high frequencies (1 GHz and higher). In fact, within loss peaks, domain wall vibrations appear as a reduction of the real part of permittivity in comparison with low frequencies (please see, e.g., data for PZT ceramics by Prof. D. Damjanovic’s group at EPFL: Appl. Phys. Lett. **94**, 212906 (2009), Appl. Phys. Lett. **96**, 242902 (2010), J. Lin’s PhD thesis at [doi:10.5075/epfl-thesis-4988](https://doi.org/10.5075/epfl-thesis-4988)). Loss peaks could be seen in principle as an enhanced conductivity in the sMIM-G channel. This means that if the observed signal from the domain walls in the sMIM-G channel would be associated with domain wall vibrations, we should see a domain wall contrast in corresponding sMIM-C images as well. However, this does not happen, which clearly points to a different mechanism of the response. Most probably, the response originating from the individual domain wall vibrations is too small to be detectable taking into account a relatively small volume of the film material involved in the response. The frequency of 3 GHz is off the loss peak (wall resonance) as well.

Finally, domain walls with a particularly high sMIM-G signal are observed around ferroelectric switching events (Fig. 2). This is evidenced from comparison with dc c-AFM, which likewise sees enhanced conductance at the domain walls created by switching (Supplementary Fig. 7a), albeit only at one bias polarity due to limitations of the dc-blocking contacts. This again proves involvement of free charge carriers in the microwave response. Please also see further arguments supporting our interpretation of the sMIM data in the response immediately below.

Reviewer #2 (Remarks to the Author):

With their revised version the authors present a detailed response to the remarks of both reviewers. They present additional evidence on the involvement of oxygen vacancies and they identify effects related to a certain roughness of the domain walls as additional contributing mechanism to the impedance. This still leaves a big question mark as to the exact microscopic mechanisms of the large impedance effect at the domain walls but including this would lead beyond the scope of this work. I thus recommend the manuscript for publication in Nature Communications. Since there is no length limit here, the authors might consider to shift some of the materials presented as supplementary information to the main text.

Response. We thank the reviewer for the high appreciation of our work. In the revised manuscript, we have moved a significant portion of Supplementary information in the main text.